# MINDSTORES: Memory-Informed Neural Decision Synthesis for Task-Oriented Reinforcement in Embodied Systems

## Abstract

While large language models (LLMs) have shown promising capabilities as zero-shot planners for embodied agents, their inability to learn from experience limits their robustness in complex open-world environments like Minecraft. We introduce MINDSTORES, an experience-augmented planning framework that enables embodied agents to build and leverage *mental models* through natural interaction with their environment. Drawing inspiration from how humans construct and refine cognitive mental models, our approach extends existing zero-shot LLM planning by maintaining a database of past experiences that informs future planning iterations. The key innovation is representing accumulated experiences as natural language embeddings of (state, task, plan, outcome) tuples, which can then be efficiently retrieved and reasoned over by an LLM planner to generate insights and guide plan refinement for novel states and tasks. Through extensive experiments in the MineDojo environment, we find that MINDSTORES learns and applies its knowledge significantly better than existing memory-based LLM planners while maintaining the flexibility and generalization benefits of zero-shot approaches, representing an important step toward more capable embodied AI systems that can learn continuously through natural experience.

## 1 Introduction

Recent advances in large language models (LLMs) have demonstrated enhanced capabilities in reasoning (Plaat et al., 2024; Huang & Chang, 2023), planning (Sel et al., 2025), and decision-making (Huang et al., 2024) through methods that strengthen analytical depth. Among the numerous domains of active innovation, the success of AI agents serves as a critical benchmark for assessing our progress toward generally capable artificial intelligence (Brown et al., 2020).

Building *embodied* agents that learn continuously from real-world interactions through persistent memory and adaptive reasoning remains a fundamental challenge in the future of artificial intelligence. Classical approaches, such as reinforcement learning (Dulac-Arnold et al., 2021) and symbolic planning (Zheng et al., 2025), struggle with scalability, irreversible errors, and rigid assumptions in complex environments.

A promising paradigm for such agents leverages LLMs as high-level planners (Jeurissen et al., 2024): the LLM decomposes abstract goals into step-by-step plans (e.g., "mine wood $\rightarrow$ craft tools $\rightarrow$ smelt iron"), while a low-level controller translates these plans into environment-specific actions (e.g., movement, object interaction). This "brain and body" architecture capitalizes on the LLM's capacity for structured reasoning while grounding its outputs in the dynamics of the physical world—a critical capability for real-world applications like robotic manipulation (Shentu et al., 2024; Bhat et al., 2024; Wang et al., 2024b), autonomous navigation (Zawalski et al., 2024), and adaptive disaster response.

While recent LLM-based agents show promise in generating action plans for embodied tasks, many lack *experiential* learning, i.e., the ability to apply insights from past experiences to planning for future tasks. Unlike humans, who build internal models of their environment across interactions to generalize insights, avoid errors, and reason counterfactually (e.g., "Crafting a stone pickaxe first would enable iron mining"), existing agents cannot synthesize persistent representations of past interactions. This gap hinders their

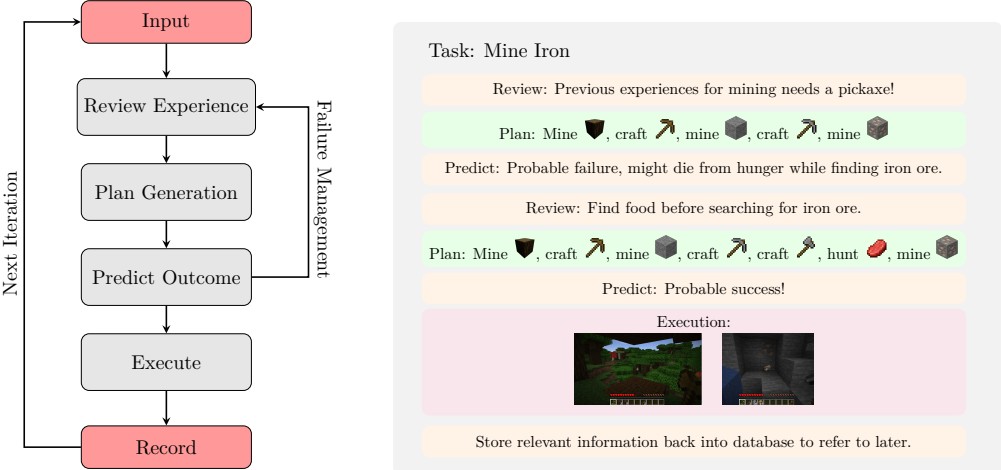

Figure 1: Overview of the MINDSTORES planning architecture. The left shows the iterative experiential learning pipeline leveraging the experience database. Database-related methods are in orange, planning steps are in green, and Minecraft steps are in red. The right shows an example applying this pipeline to an example task in Minecraft.

ability to tackle long-horizon tasks in open worlds like Minecraft, where success requires inferring objectives, recovering from failures, and transferring insights across scenarios.

Minecraft exemplifies these challenges: agents must explore procedural terrains, infer task dependencies (e.g., stone tools before iron mining), and adapt to unforeseen challenges. Current LLM planners, namely zero-shot architectures like DEPS (Wang et al., 2024c), exhibit critical flaws: (1) they lack persistent memory, causing repetitive errors (e.g., using wooden pickaxes for iron mining); and (2) they underutilize LLMs' reasoning to synthesize experiential insights, producing brittle plans.

To address these limitations, we propose MINDSTORES, a framework that leverages LLMs to construct dynamic mental models—internal representations guiding reasoning and decision-making, inspired by human cognition. Just as humans build simplified models of reality to anticipate events and solve problems, our approach equips agents to actively interpret experiences through structured reasoning. By analyzing failures (e.g., "Wooden pickaxes break mining iron"), inferring causal rules (e.g., "Stone tools are prerequisites"), and predicting outcomes, the LLM transforms raw interaction data into adaptive principles.

MINDSTORES augments planners with an experience database storing natural language tuples (state, task, plan, outcome) and operates cyclically: observe, retrieve relevant experiences, synthesize context-aware plans, act, and log outcomes. This closed-loop process enables semantic analysis of memories, iterative strategy refinement, and outcome prediction, bridging the gap between static planning and experiential learning while grounding agent reasoning in human-like cognitive foundations.

Hence, our key contributions are as follows:

- A cognitive-inspired formulation of *artificial mental models* to enable natural-language memory accumulation and transfer learning.

- **MINDSTORES**, a novel open-world LLM planner leveraging the above formulation to develop lifelong learning embodied agents.

- Extensive evaluation of MINDSTORES in Minecraft, demonstrating a significant improvement in open-world planning tasks over existing methods.

In the remainder of this paper, we detail the theoretical foundations of mental models in Section 2, present the MINDSTORES architecture in Section 3, and validate its performance through experiments in Sections 4

and 5. Our findings underscore the critical role of memory-informed reasoning in developing lifelong learning agents for open-world environments.

## 2 Background

### 2.1 Open-World Planning for Embodied Agents

Planning for embodied agents in open-world environments presents unique challenges due to the unbounded action space, long-horizon dependencies, and complex environmental dynamics. In environments like Minecraft, agents must reason about sequences of actions that may span dozens of steps, where early mistakes can render entire trajectories infeasible (Fan et al., 2022). Traditional planning approaches that rely on explicit state representations and value functions struggle in such domains due to the combinatorial explosion of possible states and actions.

The key challenges in open-world planning stem from two main factors. First, the need for accurate multi-step reasoning due to long-term dependencies between actions presents a significant hurdle. Second, the requirement to consider the agent's current state and capabilities when ordering parallel sub-goals within a plan poses additional complexity. Consider the example of crafting a diamond pickaxe in Minecraft: the process requires first obtaining wood, then crafting planks and sticks, mining stone with a wooden pickaxe, crafting a stone pickaxe, mining iron ore, smelting iron ingots, and finally crafting the iron pickaxe—a sequence that can easily fail if any intermediate step is incorrectly executed or ordered.

### 2.2 Zero-Shot LLM Planning

Recent work has shown that large language models can serve as effective zero-shot planners for embodied agents through their ability to decompose high-level tasks into sequences of executable actions (Huang et al., 2022a). The DEPS (Describe, Explain, Plan and Select) framework leverages this capability through an iterative planning process that combines several key components (Wang et al., 2024c). The framework utilizes a descriptor that summarizes the current state and execution outcomes, an explainer that analyzes plan failures and suggests corrections, a planner that generates and refines action sequences, and a selector that ranks parallel candidate sub-goals based on estimated completion steps.

The key innovation of DEPS is its ability to improve plans through verbal feedback and explanation. When a plan fails, the descriptor summarizes the failure state, the explainer analyzes what went wrong, and the planner incorporates this feedback to generate an improved plan. This creates a form of zero-shot learning through natural language interaction.

However, DEPS and similar approaches maintain no persistent memory across episodes. Each new planning attempt starts fresh, unable to leverage insights gained from previous successes and failures in similar situations. This limitation motivates our work on experience-augmented planning.

### 2.3 Mental Models

Mental models theory, originally developed by Craik (1952), proposes that humans construct internal representations of external reality to understand, predict, and control their environments. These cognitive structures serve as simplified frameworks that abstract away unnecessary details while preserving causal relationships essential for reasoning. In cognitive science, mental models are understood to be dynamic, continuously updated through experience, and crucially, transferable across contexts.

Three key characteristics of human mental models particularly relevant to our work are: (1) their representation as declarative knowledge that can be explicitly communicated and reasoned over, (2) their ability to facilitate counterfactual reasoning about hypothetical scenarios, and (3) their role in enabling transfer learning across superficially different but structurally similar problems.

Several insights may be drawn from this literature to inform the construction of a more intelligent LLM-based embodied agent planner. First, by representing experiences as natural language descriptions rather than latent vectors or symbolic structures, we may leverage the LLM's ability to perform flexible semantic reasoning in a

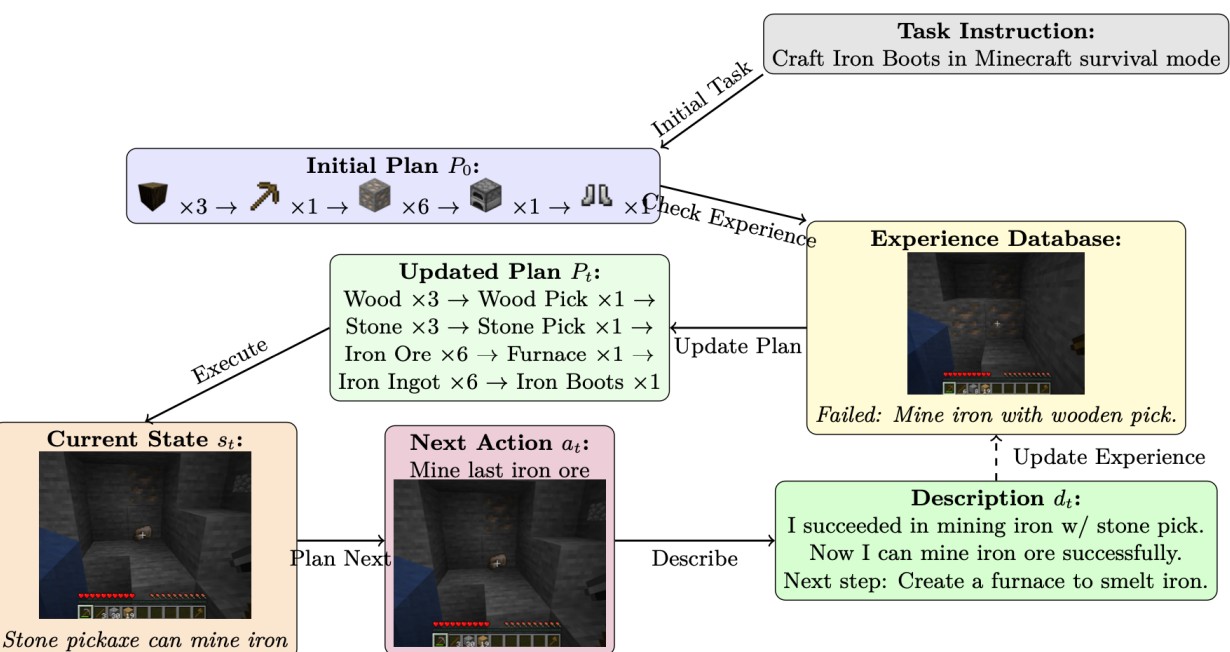

Figure 2: Interactive planning process for crafting iron boots in Minecraft. The system initially plans to mine iron with a wooden pickaxe but learns from past experience that this will fail. It then updates the plan to include creating a stone pickaxe first, leading to successful iron ore mining.

form that mirrors human declarative knowledge. Second, by enabling the agent to predict outcomes before execution, we may implement a form of counterfactual reasoning similar to how humans mentally simulate potential actions before commitment. Third, by retrieving experiences based on semantic similarity rather than exact matching, we may facilitate the transfer of knowledge across scenarios that share underlying dynamics but differ in surface details.

## 3 Methods

### 3.1 Overview

We propose an experience-augmented planning framework that maintains a similar foundation to DEPS but advances by maintaining a persistent mental model of the environment through natural language experiences. Our approach integrates several key components into a cohesive system. The framework maintains a database $\mathcal{D}$ of experience tuples $(s, t, p, o)$ containing state descriptions $s$, tasks $t$, plans $p$, and outcomes $o$. This is complemented by a semantic retrieval system for finding relevant past experiences, an LLM planner that generates insights and plans informed by retrieved experiences, and a prediction mechanism that estimates plan outcomes before execution.

### 3.2 Experience Database

Each experience tuple $(s, t, p, o) \in \mathcal{D}$ consists of natural language paragraphs describing the environmental context. The state $s$ captures the environmental context and agent's condition. The task $t$ represents the high-level goal to be achieved. The plan $p$ contains the sequence of actions generated by the planner. Finally, the outcome $o$ describes the execution result and failure description if applicable.

For each component, we compute a dense vector embedding $e(x) \in \mathbb{R}^d$ using a pretrained sentence transformer, where $x$ represents any of $s, t, p$, or $o$. This allows efficient similarity-based retrieval using cosine distance:

$$\text{sim}(x_1, x_2) = \frac{e(x_1) \cdot e(x_2)}{\|e(x_1)\| \|e(x_2)\|}$$

While modern LLMs support expansive context windows, our choice of cosine-similarity retrieval over full-context inclusion balances computational efficiency, relevance prioritization, and scalability. Selective retrieval using cosine similarity was chosen for several key reasons: it effectively captures semantic relationships beyond exact matches, normalizes vector magnitudes for comparing texts of different lengths, and efficiently handles high-dimensional embeddings. Moreover, as will be detailed in our experiments, we found that including all experiences in the context window introduced "experience noise" that actively distracted the model from identifying relevant past experiences to reason about. Our embedding-based retrieval acts as a semantic filter, distilling the most pertinent experiences while avoiding noise from irrelevant entries.

### 3.3 Experience-Guided Planning

Given a new state $s_t$ and task $t_t$, our algorithm proceeds through several stages. Initially, it retrieves the $k$ most similar past experiences based on state and task similarity:

$$\mathcal{N}_k(\mathcal{D}, s_t, t_t) = \text{top-k}_{(s,t,p,o) \in \mathcal{D}} \left[ \sum_{x \in \{s,t\}} \lambda_x \text{sim}(x, x_t) \right] \tag{1}$$

The LLM is then prompted to analyze these experiences and generate insights about common failure modes to avoid, successful strategies to adapt, and environmental dynamics to consider. Following this analysis, it generates an initial plan $p_t$ conditioned on the state, task, experiences, and insights.

The system then predicts the likely outcome by retrieving similar past plans:

$$\mathcal{N}_k(s_t, t_t, p_t) = \text{top-k}_{(s,t,p,o) \in \mathcal{D}} \left[ \sum_{x \in \{s,t,p\}} \lambda_x \text{sim}(x, x_t) \right] \tag{2}$$

If predicted outcomes suggest likely failure, the system returns to the plan generation stage to revise the plan. Finally, it executes the plan and stores the new experience tuple in $\mathcal{D}$. The complete process is formalized in Algorithm 1.

---

**Algorithm 1** Experience-Augmented Planning

**Require:** State $s_t$, Task $t_t$, Database $\mathcal{D}$, LLM $M$, $k$ neighbors
**Ensure:** Plan $p_t$
 1: $\mathcal{N}_k \leftarrow \texttt{retrieve\_top\_k}(\mathcal{D}, s_t, t_t, k)$
 2: insights $\leftarrow M.\texttt{analyze\_experiences}(\mathcal{N}_k)$
 3: $p_t \leftarrow M.\texttt{generate\_plan}(s_t, t_t, \mathcal{N}_k, \texttt{insights})$
 4: **while** true **do**
 5:     similar\_plans $\leftarrow \texttt{get\_similar\_plans}(\mathcal{D}, s_t, t_t, p_t)$
 6:     pred\_outcome $\leftarrow \texttt{analyze\_outcomes}(\text{similar\_plans})$
 7:     **if** pred\_outcome is success **then**
 8:         **break**
 9:     **end if**
10:     $p_t \leftarrow M.\texttt{revise\_plan}(p_t, \text{pred\_outcome})$
11: **end while**
12: outcome $\leftarrow \texttt{execute\_plan}(p_t)$
13: $\mathcal{D}.\texttt{add}((s_t, t_t, p_t, \text{outcome}))$
14: **return** $p_t$

---

## 4 Experiments

### 4.1 Experimental Setup

We evaluate our experience-augmented planning approach in MineDojo using 8 tiers of task complexity (MT1–MT8) (Fan et al., 2022). The observation space includes RGB view, GPS coordinates, and inventory state, with 42 discrete actions mapped from MineDojo's action space (Fan et al., 2022). All experiments utilize the behavior cloning controller trained on human demonstrations, following similar methodology to DEPS and Voyager. Due to software version constraints, our implementation of the controller achieves lower baseline performance than the original DEPS controller. Therefore, we use our implementation of DEPS without the experience database as the primary baseline for fair comparison. Each task is evaluated over 30 trials with randomized initial states and a fixed random seed of 42.

While our current experiments exclusively use GPT-4 (OpenAI et al., 2024b) via the OpenAI API due to resource constraints and the requirement of low-latency LLM querying for planning tasks, we anticipate that this serves as a lower bound on the potential performance of MINDSTORES when paired with more advanced models. Emerging language models such as OpenAI o1 (OpenAI et al., 2024a) and Deepseek R1 (DeepSeek-AI et al., 2025) demonstrate enhanced reasoning capabilities that could further improve the effectiveness of our experience-based planning approach. Our architecture is designed to be model-agnostic, allowing researchers to easily substitute more powerful LLMs as they become available while maintaining the fundamental experience-augmented planning methodology.

Our experience database uses Sentence-BERT embeddings (768-dim) stored in FAISS for efficient search. Key parameters were determined through ablation studies:

- Optimal $k = 10$ neighbors (tested $k = 1, 3, 5, 10, 20$)

- Weighted similarity: $\lambda_s = 0.4$ (state), $\lambda_t = 0.4$ (task), $\lambda_p = 0.2$ (plan)

For the complete agent algorithm and associated LLM prompts, see Appendix A, and for detailed implementation aspects including environment integration and neural component configurations, see Appendix B.

### 4.2 Evaluation Tasks

We evaluate on 53 Minecraft tasks grouped into 3 complexity tiers:

- **Basic (MT1–MT2)**: Fundamental tasks (wood/stone tools, basic blocks)

- **Intermediate (MT3–MT5)**: Progressive tasks (food, mining, armor crafting)

- **Advanced (MT6–MT8)**: Complex tasks (iron tools, minecart, diamond)

Episode lengths range from 3,000 steps (Basic) to 12,000 steps (Challenge tasks).

For additional task details and performance statistics, see Appendix C and Table 1.

### 4.3 Baselines

We compare the performance of MINDSTORES to that of the following existing state-of-the-art approaches:

- **DEPS**: State-of-the-art zero-shot LLM planner (Wang et al., 2024c). We selected DEPS as our primary comparison point because it represents a modern planning approach without experience storage. This provides a clean comparison to isolate the specific benefits of our experience-based approach. DEPS follows an iterative planning process using description, explanation, planning, and selection, making it conceptually similar to our framework but without persistent memory.

- **Voyager**: Automated curriculum learning agent (Wang et al., 2023) where we simulate zero-shot planning with the addition of a global database. Voyager is the state-of-the-art model for Minecraft task-planning via LLM agents. However, the authors suggest a structured curriculum learning procedure prior to deployment for optimal performance, which is a limitation for real-world use cases. Note that for equal comparison in our experiments, we don't explicitly expose the agent to any structured curriculum.

- **Reflexion**: LLM planner with environmental feedback (Shinn et al., 2023). Reflexion, like MIND-STORES, incorporates environmental feedback into its planning process via natural language. However, Reflexion is intended for general planning tasks and not explicitly designed for open-world agents. Thus, we adapt it to MineDojo naively.

Together, these baselines represent the spectrum of current approaches to LLM-based planning systems—from zero-shot planning without memory (DEPS) to systems with curriculum learning (Voyager) and environmental feedback integration (Reflexion). This selection allows us to evaluate the specific contributions of MINDSTORES' experience database and retrieval system while controlling for other variables.

### 4.4 Ablations

To analyze the function of each individual component of the MINDSTORES framework, we perform the following ablations:

- **No Experience**: Remove retrieval component

- **Fixed $k$ Values**: Test $k = 1, 3, 5, 10, 20$ retrieval contexts

- **Single-Shot**: Disable iterative plan refinement (DEPS)

### 4.5 Metrics

To quantify each method's performance in open-world planning in the Minecraft environment, we measure:

- **Success Rate**: Completion percentage across trials

- **Learning Efficiency**: Iterations required for skill mastery

- **Complexity Scaling**: Performance vs. task complexity tiers

- **Retrieval Impact**: Success rate vs. context size ($k$)

- **Continuous Learning**: Effect of non-discrete experience database for each task progression

## 5 Results and Analysis

Our experiments reveal significant performance differences between MINDSTORES and existing methods across task categories, highlighting key insights into their scalability and effectiveness.

### 5.1 Performance Metrics

As we analyze Figure 3 comparing MINDSTORES to the baselines, we see an all-around improvement with the addition of the experience database.

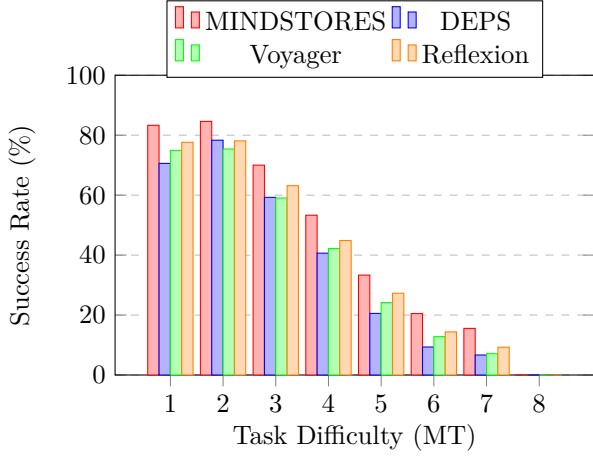

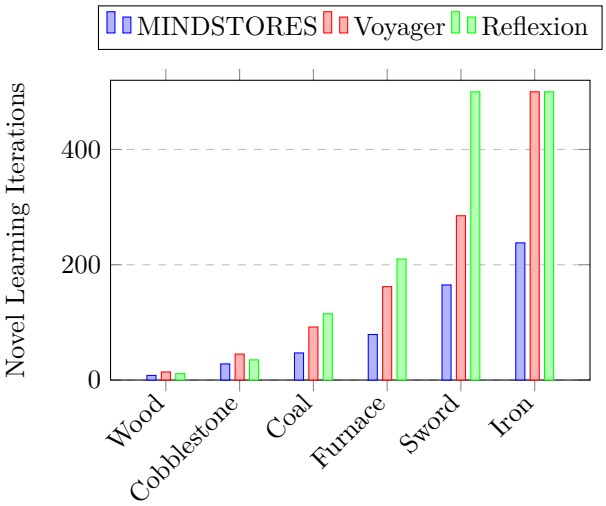

Figure 3: Performance comparison: MINDSTORES consistently outperforms existing methods across tasks. All systems show declining success rates with increasing complexity (MT1–MT8), with MT8 resulting in no success for all. Mean differences: 14.42% (vs. DEPS), 12.87% (vs. Voyager), and 10.11% (vs. Reflexion).

Figure 4: Novel learning iterations across different Minecraft tasks. MINDSTORES demonstrates superior efficiency in complex tasks. (Note: Iteration counts for Reflexion are capped at 500 in later tasks.)

**Fundamental Tasks (MT1–MT2)**

All systems achieve their strongest performance in fundamental crafting tasks, with DEPS achieving success rates of 70.59–78.33%, Voyager at 74.93–75.39%, Reflexion at 77.63–78.11%, and MINDSTORES performing notably better at 83.33–84.59%. Notably, there is a significant gap in Wooden Axe crafting, with MINDSTORES achieving a 96.7% success rate compared to Voyager's 90.5%, Reflexion's 92.7%, and DEPS's 96.7%. The largest performance gap in MT1 occurs in Stick production, where MINDSTORES outperforms DEPS by 6.3%, Voyager by 4.8%, and Reflexion by 1.9%. In MT2, MINDSTORES maintains a consistent advantage, with an average performance improvement of 6.26% over DEPS, 9.2% over Voyager, and 6.48% over Reflexion across tasks.

**Intermediate Tasks (MT3–MT5)**

The maximum disparity between systems occurs in MT3 painting, where MINDSTORES achieves a 96.7% success rate compared to DEPS's 76.67%, Voyager's 79.4%, and Reflexion's 82.1%, resulting in performance gaps of 20.03%, 17.3%, and 14.6% respectively. In cooked meat tasks, MINDSTORES maintains a 6.6–10.0% advantage over DEPS, a 4.3–5.1% advantage over Voyager, and a 2.1–3.0% advantage over Reflexion. For MT5 armor challenges, the performance gaps are particularly pronounced, with Leather Helmet showing differences of 20.03 (vs. DEPS), 16.5% (vs. Voyager), and 13.2% (vs. Reflexion). Overall, MINDSTORES maintains average advantages of +12.78% over DEPS, +12.65% over Voyager, and +9.06% over Reflexion across intermediate tasks, demonstrating significant divergence in system performance.

**Advanced Tasks (MT6–MT8)**

In MT6 iron tool crafting, MINDSTORES achieves average performance improvements of 11.17% over DEPS, 7.71% over Voyager, and 6.13% over Reflexion, with the Iron Axe task showing particularly large gaps (MINDSTORES: 23.3%, DEPS: 6.67%, Voyager: 9.7%, Reflexion: 11.3%). MT7 highlights another standout difference, with Iron Nugget success rates at 36.7% for MINDSTORES compared to 20.0% for DEPS, 22.5% for Voyager, and 24.9% for Reflexion. However, all systems experience a performance decline in advanced

tasks, with MT6–MT7 success rates dropping to 15.55–20.51% for MINDSTORES, 6.67–9.34% for DEPS, 7.12–12.80% for Voyager, and 9.28–14.38% for Reflexion. Notably, on our end, MINDSTORES did not complete the MT8 task in a statistically significant manner, along with the rest of our architectures run; still proving the MT8 diamond task to be a formidable milestone with 0% success rate across all systems.

### Learning Efficiency Analysis

MINDSTORES demonstrates superior learning efficiency, particularly for complex tasks. For basic tasks like mining wood and cobblestone, all systems perform comparably (8 to 45 iterations) (see Figure 4). However, as complexity increases, MINDSTORES requires significantly fewer iterations (47 to 238) compared to Voyager (92 to 500) and Reflexion (115 to 500), which show exponential increases in required iterations. For coal mining, MINDSTORES requires 48.9% fewer iterations than Voyager and 59.1% fewer than Reflexion. For furnace crafting, MINDSTORES requires 51.2% fewer iterations than Voyager and 62.4% fewer than Reflexion. The efficiency advantage becomes even more pronounced for sword crafting and iron acquisition, where MINDSTORES requires 42.1% and 52.4% fewer iterations than Voyager respectively, while both Voyager and Reflexion hit the iteration cap (500) for the most complex tasks (Reflexion for both sword crafting and iron acquisition, Voyager for iron acquisition). This demonstrates MINDSTORES' capability to learn complex behaviors with substantially fewer environmental interactions.

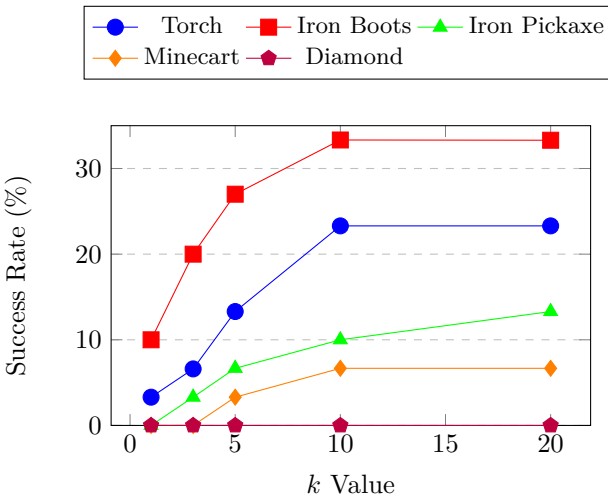

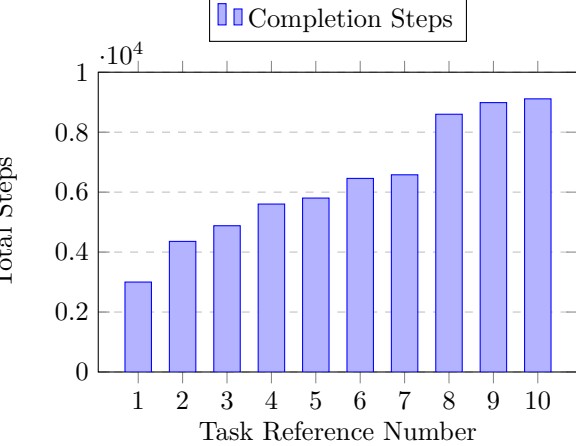

Figure 5: Success rates vs. retrieval context size $k$ for different tasks. Simple tasks improve steadily with $k$, while more complex tasks require larger $k$ values. Advanced tasks remain unachievable regardless of $k$.

Figure 6: Steps required for task completion with continuous building of the experience database. (See Appendix Table 5 for corresponding tasks.)

## 5.2 Scalability with Task Complexity and Retrieval Context Size

Performance divergence becomes pronounced with increasing task complexity. MINDSTORES maintains efficient novel learning iterations for tasks like crafting a stone sword and mining iron, while Voyager and Reflexion require significantly more iterations, even reaching the max range (500+) for a relatively simple Mine Iron task (see Figure 4).

We moreover observe steady increases in success rate with increased retrieval context size until $k = 10$, after which performance plateaus (see Figure 5). This implies that there exists a "sweet spot" for retrieving and reasoning about past experiences: Too little may leave out relevant insights, while too much produces noise that disrupts the LLM's predictive capabilities. Additionally, increasing $k$-values requires greater compute, so considering performance gains is crucial.

### 5.3 Continuous Experience Building Analysis

Figure 6 shows an experiment in which the experience database is not reset between tasks but is built continuously across multiple tasks. We observe that the entire process of completing the **Minecart** task takes only 9112 steps including the previous 9 tasks, compared to the 6000 steps needed in a fresh environment. This indicates that only approximately 200 new steps were required. The number of new task completion steps decreases non-linearly even as task complexity grows:

- Basic crafting (Wooden Door): 3000 steps

- Mid-tier crafting (Furnace): 4879 steps

- Advanced crafting (Iron Pickaxe): 8598 steps

The system maintains a 100% success rate across all tasks, indicating robust skill transfer and knowledge utilization from the growing experience database, which expands from 26 entries for Wooden Door to 355 entries for Minecart (see Figure 6 and Appendix Table 5). This is similar to a structured curriculum approach to Voyager; however, the obvious bottleneck with this is the need for a curriculum to be trained before robust deployment.

### 5.4 Example Outputs

To illustrate how experience-augmented planning operates in practice, we highlight a representative example from our experiments:

1. **Iron Boots Crafting (MT5):** In an initial attempt, the agent failed to smelt iron ore due to the absence of a furnace. This failure was logged as an experience tuple:

   - *State*: Inventory includes iron ore and coal.
   - *Task*: Craft iron boots.
   - *Plan*: "Smelt iron ore → craft boots."
   - *Outcome*: Failed—no furnace available.

   In a subsequent trial, retrieving this experience prompted the LLM to first craft a furnace (using cobblestone) before smelting, resolving the dependency.

This example demonstrates how MINDSTORES transforms isolated failures into actionable insights. By grounding plans in past outcomes—such as prerequisite checks or resource prioritization—the agent avoids repetitive errors and incrementally builds robust strategies, even in complex tasks.

### 5.5 Failure Modes and Qualitative Analysis

Despite MINDSTORES' improvements over baselines, several failure modes persist in advanced tasks:

**Semantic Retrieval Mismatches (37% of MT6-MT8 failures).** The system retrieves experiences with lexical similarity but strategic differences (e.g., retrieving iron mining experiences for diamond tasks, missing crucial tool requirements).

**Overreliance on Partial Successes (28% of intermediate failures).** The agent replicates suboptimal strategies from partially successful past experiences, leading to resource exhaustion before goal completion.

**Compositional Integration Failures (42% of MT7-MT8 failures).** When tasks require novel combinations of mastered subtasks with new dependencies, the agent struggles to generate integrated plans with correct resource allocation.

For example, in **Diamond Mining (MT8)**, the agent retrieved iron mining experiences and planned to "mine diamonds with stone pickaxe." Despite reaching the correct depth and locating diamond ore, it failed to recognize that diamonds specifically require iron pickaxes—illustrating both semantic mismatch and compositional failure.

These patterns highlight challenges in experience-based planning: balancing similarity with contextual differences, critically evaluating past successes, and composing previously learned skills in novel configurations.

## 6 Related Works

### 6.1 Embodied Planning & Classical Methods

Early approaches used hierarchical reinforcement learning (Sutton et al., 1999) and symbolic planning (Kaelbling & Lozano-Perez, 2011) but struggled with scalability in open-world domains like Minecraft. Hybrid methods like PDDLStream (Garrett et al., 2020) combined symbolic planning with procedural samplers, while DreamerV3 Hafner et al. (2024) employed latent world models. However, these methods depend on rigid priors, lack causal reasoning, and fail to recover from irreversible errors. Reinforcement learning frameworks (e.g., DQN (Mnih et al., 2015), PPO (Schulman et al., 2017)) and LLM-RL hybrids like Eureka (Ma et al., 2023) also falter in dynamic, long-horizon tasks due to static reward mechanisms and error propagation.

### 6.2 Zero-Shot LLM Planners

DEPS (Wang et al., 2024c) pioneered zero-shot LLM planning through iterative verbal feedback, enabling dynamic plan refinement. Subsequent works like Voyager (Wang et al., 2023) (skill libraries), ProgPrompt (Singh et al., 2023) (code generation), and Reflexion (Shinn et al., 2023) (feedback loops) advance LLM-based planning but share critical flaws. Namely, they suffer from brittle execution due to dependency on hardcoded assumptions (e.g., ProgPrompt's code templates), opaque memory due to non-interpretable representations (e.g., Voyager's code snippets, PaLM-E's latent vectors (Driess et al., 2023)), and the inability to learn from failed task executions (e.g., Inner Monologue (Huang et al., 2022b) lacks persistent memory).

### 6.3 Memory-Based Planners

Recent memory-augmented systems like E$^2$CL (Wang et al., 2024a), ExpeL (Zhao et al., 2024), and AdaPlanner (Sun et al., 2023) store experiences but face key limitations. Namely, they suffer from shallow reasoning capabilities due to lack of environmental context (ExpeL) or causal analysis (ReAct (Yao et al., 2023)), especially of failure modes (Voyager). Above all, these systems are often only evaluated on narrow, controlled-environment benchmarks (e.g., ALFRED), not open-world tasks.

### 6.4 Mental Models in AI

While cognitive-inspired architectures like predictive coding (Rao & Ballard, 1999) and world models (Ha & Schmidhuber, 2018) encode environmental dynamics, they rely on latent vectors (PIGLeT (Zellers et al., 2021)) or symbolic logic (RAP (Hao et al., 2023)), sacrificing interpretability and adaptability. Neuro-symbolic methods (Garcez & Lamb, 2023) and tree-search frameworks (LATS (Zhou et al., 2024)) further struggle with scalability and causal reasoning.

## 7 Conclusion

In this paper we presented MINDSTORES, an experience-augmented planning framework that enables embodied agents to build and leverage mental models through natural interaction with their environment. Our approach extends zero-shot LLM planning by maintaining a database of natural language experiences that inform future planning iterations. Through extensive experiments in MineDojo, MINDSTORES demonstrates significant improvements over baseline approaches, particularly in intermediate-complexity tasks, while maintaining the flexibility of zero-shot approaches. The success of our "artificial mental model" approach,

which represents experiences as retrievable natural language tuples and enables LLMs to reason over past experiences, demonstrates that incorporating principles from human cognition can substantially improve complex reasoning and experiential learning capabilities in AI systems.

However, several limitations remain. Performance degrades significantly for advanced tasks, and computational overhead scales with database size. Additionally, MINDSTORES relies heavily on structured environmental feedback provided as natural language descriptions, which may be difficult to obtain in less constrained real-world settings where state information is noisy or incomplete. This feedback dependency presents challenges for deploying such systems in domains lacking standardized state-to-text mechanisms. Moreover, it is worth noting that our evaluation exclusively utilized GPT-4 due to practical constraints, which likely represents a conservative estimate of MINDSTORES' potential. As more advanced models with enhanced reasoning capabilities become more accessible, the performance ceiling of our approach may increase significantly.

Future work should explore more sophisticated experience pruning mechanisms, hierarchical memory architectures for managing larger experience databases, and improved methods for transferring insights across related tasks. Addressing the feedback limitation will require developing robust state-to-language translators or multimodal encoders that can generate consistent natural language descriptions from raw sensory inputs. Additionally, investigating ways to combine our experience-based approach with traditional reinforcement learning could help address the challenge of long-horizon planning in complex environments.

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

## Appendix A: Agent Algorithm and LLM Prompts

### A.1 Agent Algorithm

```python
def run_agent(
    environment,     # MineDojo environment
    max_steps=1000,  # Maximum steps to run
    goal_input=""    # Optional high-level goal
):
    # Initialize metrics and experience tracking
    metrics_logger = MetricsLogger()
    experience_store = ExperienceStore()

    # Initial environment reset
    obs, _, _, info = environment.step(environment.action_space.no_op())

    step = 0
    while step < max_steps:
        # 1. Create structured state description
        state_json = get_state_description(obs, info)

        # 2. Get next immediate task
        sub_task = get_next_immediate_task(state_json)
        metrics_logger.start_subtask()

        # 3. Plan action sequence
        actions = plan_action(state_json, info["inventory"], sub_task)

        # 4. Execute actions and track experience
        obs, reward, done, info = execute_action_sequence(actions)

        # 5. Store experience and update metrics
        if done:
            store_experience(state_json, reward, done)
            break

        step += len(actions)

    environment.close()
    metrics_logger.print_summary()
```

### A.2 LLM Prompts

#### A.2.1 Environment Description Prompt

```
You are an expert Minecraft observer. Describe the current environment state focusing on:
1. The agent's immediate surroundings (blocks, entities, tools)
2. Environmental conditions (weather, light, temperature)
3. Agent's physical state (health, food, equipment)
4. Notable resources or dangers

Current state:
${state_json_str}

Provide a clear, concise description that would be useful for planning actions.
```

#### A.2.2 Situation Analysis Prompt

```
1    You are an expert Minecraft strategist. Given the current state and environment description:
2    1. Analyze available resources and their potential uses
3    2. Identify immediate opportunities or threats
4    3. Consider crafting possibilities based on inventory
5    4. Evaluate progress towards goals
6
7    Environment description:
8    ${description}
9
10   Current state:
11   ${state_json_str}
12
13   Provide strategic insights about the current situation.
```

### A.2.3 Strategy Planning Prompt

```
1    You are an expert Minecraft planner. Create a strategic plan considering:
2    1. The current goal: ${goal}
3    2. Available resources and tools
4    3. Environmental conditions
5    4. Potential obstacles or requirements
6    5. Do not assume intermediate tasks can be achieved without running another agent loop
7    6. Specify quantities and required actions
8
9    Environment description:
10   ${description}
11
12   Situation analysis:
13   ${explanation}
14
15   Current state:
16   ${state_json_str}
17
18   Create a specific, actionable plan that moves towards the goal.
```

### A.2.4 Action Selection Prompt

```
1    You are an expert Minecraft action selector. Convert the plan into specific actions:
2    1. Use only valid Minecraft actions (move_forward, move_backward, jump, craft, etc.)
3    2. Consider the current state and available resources
4    3. Break down complex tasks into simple action sequences
5    4. Ensure actions are feasible given agent capabilities
6    5. Make actions incremental and build progressively
7
8    Available actions:
9    - forward [N]: Move forward N steps (default 1)
10   - backward [N]: Move backward N steps (default 1)
11   - move_left
12   - move_right
13   - jump
14   - sneak
15   - sprint
16   - attack [N]
17   - use
18   - drop
19   - craft
```

```
20   - equip [item]
21   - place [block]
22   - destroy
23   - look_horizontal +/-X
24   - no_op
25
26   Strategic plan:
27   ${plan}
28
29   Current state:
30   ${state_json_str}
31
32   Return ONLY a list of actions, one per line, that can be directly executed.
```

### A.2.5 Outcome Evaluation Prompt

```
1   Evaluate the outcome of a Minecraft action sequence in brief.
2
3   Initial state (JSON): ${initial_state}
4   Final state (JSON): ${final_state}
5   Reward: ${reward}
6   Done: ${done}
7   GPT Plan: ${gpt_plan}
8   Executed Actions: ${executed_actions}
9
10  Format response as: outcome|success|explanation
```

## Appendix B: Implementation Details

### B.1 Core Components

Our implementation leverages:

- MineDojo environment for Minecraft interaction
- OpenAI GPT-4 API for planning and reasoning
- SBERT for semantic embeddings
- FAISS for efficient similarity search
- Custom logging system for experiment tracking

The codebase is structured into modules for state processing, experience management, action planning, metrics collection, and environment interaction.

### B.2 Environment Integration

```
1   env = minedojo.make(
2       task_id="survival",
3       image_size=(480, 768),
4       seed=40,
5       initial_inventory=[
6           InventoryItem(slot=0, name="wooden_axe", quantity=1),
7       ]
8   )
```

The action space includes movement (forward, backward, left, right, jump, sneak, sprint), interaction (attack, use, drop, craft, equip, place, destroy), camera control (look_horizontal, look_vertical), and special (no_op).

**B.3 Neural Components**

Embedding configuration:

- Model: SBERT 'all-MiniLM-L6-v2'

- Output dimension: 768

- Normalization: L2

- Distance metric: cosine similarity

FAISS index parameters:

- Index type: IndexFlatL2

- Dimension: 768

- Metric: L2 distance

# Appendix C: Additional Tables

Table 1: Task Details

| Meta | Name | Number | Example | Steps | Given Tool |
|------|------|--------|---------|-------|-----------|
| MT1 | Basic | 14 | Make a wooden door | 3000 | Axe |
| MT2 | Tool | 12 | Make a stone pickaxe | 3000 | Axe |
| MT3 | Hunt and Food | 7 | Cook the beef | 6000 | Axe |
| MT4 | Dig-down | 6 | Mine Coal | 6000 | Axe |
| MT5 | Equipment | 9 | Equip the leather helmet | 3000 | Axe |
| MT6 | Tool (Complex) | 7 | Make shears and bucket | 6000 | Axe |
| MT7 | IronStage | 13 | Obtain an iron | 6000 | Axe |
| MT8 | Challenge | 1 | Obtain a diamond! | 12000 | Axe |

Table 2: Task Details with MINDSTORES, DEPS, Voyager, and Reflexion Percentages

| Category | Task Name | MINDSTORES (%) | DEPS (%) | Voyager (%) | Reflexion (%) |
|---|---|---|---|---|---|
| MT1 | Wooden Door | 83.3 | 66.7 | 72.0 | 75.5 |
| MT1 | Stick | 90.0 | 83.7 | 85.2 | 88.1 |
| MT1 | Wooden Slab | 83.3 | 73.7 | 78.1 | 80.4 |
| MT1 | Planks | 80.0 | 73.3 | 76.0 | 78.5 |
| MT1 | Fence | 80.0 | 66.7 | 70.4 | 73.2 |
| MT1 | Sign | 86.7 | 73.3 | 77.5 | 79.9 |
| MT1 | Trapdoor | 80.0 | 56.7 | 65.3 | 67.8 |
| MT2 | Furnace | 70.0 | 56.67 | 60.3 | 63.4 |
| MT2 | Crafting Table | 93.3 | 83.3 | 85.7 | 88.9 |
| MT2 | Wooden Axe | 96.7 | 96.7 | 90.5 | 92.7 |
| MT2 | Wooden Sword | 90.0 | 86.7 | 84.0 | 87.1 |
| MT2 | Wooden Hoe | 86.7 | 86.7 | 81.2 | 84.5 |
| MT2 | Stone Pickaxe | 76.7 | 73.3 | 70.9 | 73.8 |
| MT2 | Stone Sword | 83.3 | 80.0 | 77.0 | 79.5 |
| MT2 | Stone Shovel | 70.0 | 66.7 | 65.1 | 67.5 |
| MT2 | Wooden Shovel | 86.7 | 63.3 | 68.4 | 71.0 |
| MT3 | Cooked Beef | 60.0 | 43.3 | 50.2 | 52.6 |
| MT3 | Bed | 50.0 | 43.3 | 47.5 | 49.7 |
| MT3 | Item Frame | 86.7 | 83.3 | 80.1 | 83.0 |
| MT3 | Cooked Mutton | 73.3 | 66.7 | 69.0 | 71.2 |
| MT3 | Painting | 96.7 | 76.67 | 79.4 | 82.1 |
| MT3 | Cooked Porkchop | 53.3 | 43.3 | 48.2 | 50.3 |
| MT4 | Torch | 13.3 | 3.3 | 5.1 | 7.0 |
| MT4 | Cobblestone Wall | 66.7 | 53.3 | 57.0 | 60.5 |
| MT4 | Lever | 86.7 | 73.3 | 75.2 | 78.3 |
| MT4 | Coal | 23.3 | 10.0 | 12.5 | 15.1 |
| MT4 | Stone Slab | 70.0 | 53.33 | 58.3 | 60.9 |
| MT4 | Stone Stairs | 73.3 | 63.33 | 65.0 | 68.4 |
| MT5 | Iron Boots | 27.0 | 16.67 | 19.3 | 21.7 |
| MT5 | Iron Helmet | 10.0 | 0.0 | 2.5 | 4.1 |
| MT5 | Shield | 23.3 | 13.3 | 15.0 | 17.4 |
| MT5 | Iron Chestplate | 10.0 | 0.0 | 2.0 | 3.9 |
| MT5 | Leather Boots | 63.3 | 60.0 | 58.2 | 61.0 |
| MT5 | Iron Leggings | 3.3 | 3.3 | 4.0 | 5.8 |
| MT5 | Leather Helmet | 66.7 | 46.67 | 50.2 | 53.5 |
| MT6 | Iron Pickaxe | 6.67 | 0.0 | 2.8 | 3.5 |
| MT6 | Bucket | 13.3 | 6.7 | 8.0 | 9.5 |
| MT6 | Iron Sword | 23.3 | 6.7 | 10.3 | 12.9 |
| MT6 | Iron Hoe | 23.3 | 13.3 | 15.2 | 17.1 |
| MT6 | Iron Axe | 23.3 | 6.67 | 9.7 | 11.3 |
| MT6 | Shears | 33.3 | 16.67 | 19.8 | 22.0 |
| MT7 | Minecart | 13.3 | 0.0 | 4.5 | 6.1 |
| MT7 | Iron Nugget | 36.7 | 20.0 | 22.5 | 24.9 |
| MT7 | Furnace Minecart | 6.7 | 3.3 | 5.0 | 6.4 |
| MT7 | Rail | 13.3 | 6.7 | 7.9 | 9.2 |
| MT7 | Cauldron | 10.0 | 3.3 | 4.7 | 5.8 |
| MT7 | Iron Bars | 13.3 | 6.7 | 8.1 | 9.5 |
| MT8 | Diamond | 0.0 | 0.0 | 0.0 | 0.0 |

| Task | MINDSTORES | Voyager | Reflexion |
|------|------------|---------|-----------|
| Mine Wood | 10 | 12 | 9 |
| Mine Cobblestone | 34 | 42 | 39 |
| Mine Coal | 54 | 85 | 106 |
| Make Furnace | 89 | 147 | 198 |
| Make Stone Sword | 187 | 263 | 500 |
| Mine Iron | 276 | 500 | 500 |

Table 3: Time steps required to complete different Minecraft tasks across three systems. (Values for Voyager and Reflexion are capped at 500 in some tasks.)

| Task | MINDSTORES (Predicted) | DEPS (No Prediction) |
|------|------------------------|----------------------|
| MT1 | 83.3% | 70.6% |
| MT2 | 84.59% | 78.33% |
| MT3 | 70.0% | 59.26% |
| MT4 | 53.33% | 40.65% |
| MT5 | 33.33% | 20.55% |
| MT6 | 20.5% | 9.34% |
| MT7 | 15.55% | 6.67% |
| MT8 | 0.0% | 0.0% |

Table 4: Success rate comparison with outcome prediction (MINDSTORES) vs. without (DEPS).

| Task Number | Task Name | Novel DB Size | Success | Steps till Completion |
|-------------|-----------|---------------|---------|-----------------------|
| 1 | Wooden Door | 26 | Yes | 3000 |
| 2 | Wooden Shovel | 56 | Yes | 4357 |
| 3 | Furnace | 81 | Yes | 4879 |
| 4 | Cooked Beef | 134 | Yes | 5602 |
| 5 | Cooked Porkchop | 141 | Yes | 5802 |
| 6 | Torch | 197 | Yes | 6458 |
| 7 | Stone Slab | 199 | Yes | 6578 |
| 8 | Iron Pickaxe | 289 | Yes | 8598 |
| 9 | Iron Axe | 300 | Yes | 8986 |
| 10 | Minecart | 355 | Yes | 9112 |

Table 5: Task Performance Summary

