# OpenReview forum: "MINDSTORES: Memory-Informed Neural Decision Synthesis for Task-Oriented Reinforcement in Embodied Systems"
_TMLR — Rejected by TMLR_

### Review · Reviewer_ctoN · 2025-02-28

**Summary Of Contributions:**

This paper presents MindStores, an LLM-agent that stores experiences in language tuples and retrieves them to generate better plans. Unlike reactive agents that map observations to actions, MindStores saves past experiences as language tuples (states, tasks, plans, outcomes), and retrieves the top K relevant experiences (based on cosine similarity) before solving a task. These past experiences are iterated with an LLM to analyze past failures and propose alternative action plans. MindStores is evaluated in the Minedojo environment on 8 tiers of task complexity. These tasks range from simple tasks like wood-axe crafting to complex and long-horizon tasks like mining diamonds. MindStores is benchmarked against state-of-the-art LLM-agents like DEPS, Voyager, and Reflexion. MindStores achieves 9.4% mean improvement over DEPS.

**Audience:**

Yes

**Broader Impact Concerns:**

No major concerns.

**Claims And Evidence:**

No

**Requested Changes:**

See weaknesses above. To summarize:
- Add missing results.
- Justify the choice of baselines.
- Evaluate MindStores on more domains.
- Is cosine-similarity retrieval the best way to find experiences?
- Discuss failure modes and qualitative cases.

**Strengths And Weaknesses:**

Strengths:

- MindStores tackles the important and challenging problem of learning from past experiences. Most agents reactively map observations to actions, and even recent LLM-agents that learn from failures react within a short experience horizon. Retrieving past experiences to learn from failures and generate alternative action plans seems like a general and powerful solution to improve long-horizon reasoning capabilities of agents.
- MindStores’ approach is simple and easy to understand. Experiences are stored as language tuples (states, tasks, plans, outcomes) and retrieved with cosine-similarity from language embeddings.
- The appendix includes prompts and detailed descriptions of the algorithm. These details should make it easy to reproduce the agent. Since the evaluations are also conducted on an open-source benchmark (Minedojo), the experimental results should be easily reproducible.
- MindStores achieve a 9.4% mean improvement over DEPS (Describe, Explain, Plan, Select), a state-of-the-art LLM-agent.
The experiments try to study MindStores with a wide range of metrics like success rate, learning efficiency, complexity scaling, retrieval size, and continual learning.

Weaknesses:
- Key concern: the results in Section 5 seem half-baked and incomplete. More specifically:
(1) Section 4.3 mentions three baselines: DEPS, Voyager, and Reflexion. But in Figure 3, only DEPS is reported with success rates. How do Voyager and Reflexion perform in terms of success rates?
(2) Section 4.4 mentions ablations like “No Experiences” and “Single-Shot DEPS”. But Figure 5 only reports a sensitivity analysis on k values. The appendix doesn’t seem to mention these results as well.
(3) In general, the experimental setup in Section 4 seems disconnected from the actual results in Section 5. Why were these results omitted? Or is this an incomplete submission?
- The choice of the three baselines (DEPS, Voyager, Reflexion) needs to be justified. These three baselines were designed for different purposes: DEPS is an LLM planner, Voyager is curriculum agent, and Reflexion incorporates environmental feedback. Reflexion seems the most relevant since it also learns from environmental feedback. But if so, why was DEPS, a pure LLM-planner, chosen as the main baseline in Section 5?
- The experiments are limited to a single domain: Minedojo. While the paper understandably focuses on Minecraft as the main test-bed, MindStores itself doesn’t seem limited to Minedojo. DEPS and Reflexion are evaluated in several domains like robotic manipulation and simulated household tasks (ALFWorld). MindStores’ results can be greatly strengthened if it were demonstrated that MindStore achieves compelling gains across several domains.
- The cosine-similarity based retrieval seems like “a solution”, but not “the solution” for finding relevant experiences. Modern LLMs can handle very long context lengths like 128k+ tokens (or even 1M like Gemini). Since the language tuples are quite simple, would it be possible to put all experiences in the context?
- The discussion or results section could benefit from an analysis on failure modes. What causes the retrieval based planning to fail? What are some common failure modes? Understanding these cases would greatly help in figuring out the capabilities and shortcomings of the agent.
- A key limitation of MindStore is that it relies on good environmental feedback which is provided as natural language descriptions. This assumption works for Minecraft, but doesn’t translate to true open-world scenarios in real-world settings.
- The results section or appendix could benefit from qualitative cases where retrieving an old experience helped solve a difficult task. Quantitative metrics are great, but it would be nice to know what experience-augmented planning entails in practice.

---

> ### Author Response · Authors · 2025-04-02
> **Official Comment by Authors**
>
> Thank you for your thoughtful feedback and detailed review. We appreciate your recognition of MindStores' strengths in tackling the challenging problem of learning from past experiences, its simple yet powerful approach, and the comprehensive documentation in the appendix. We address your concerns below:
>
> **Missing Results and Experimental Inconsistencies**
>
> We sincerely apologize for the confusion caused by our submission. You are completely correct that some results were unintentionally omitted from the initial submission. This was an unfortunate oversight during the compilation of our manuscript. We have uploaded the complete results.
>
> **Choice of Baselines**
>
> Thank you for highlighting the need to better justify our baseline selection. We selected these three baselines to represent different approaches to LLM-based agents:
>
> DEPS was chosen as our primary comparison point because it represents a state-of-the-art planning approach without experience storage. This provides a clean comparison to isolate the specific benefits of our experience-based approach.
> Reflexion incorporates environmental feedback but lacks the structured experience storage and retrieval mechanism of MindStores.
> Voyager utilizes curriculum learning, which complements our evaluation of how agents handle tasks of increasing complexity.
> In the revised manuscript, we will expand this justification and provide a more thorough comparative analysis of all three baselines.
>
> **Evaluation Across Multiple Domains**
>
> Thank you for your insightful suggestion regarding evaluating MindStores across multiple domains. We acknowledge that our current evaluation is limited to the Minedojo environment.
>
> We chose to focus on Minecraft because it serves as a uniquely challenging and representative environment for testing embodied agent capabilities. Minecraft requires long-horizon planning (crafting sequences spanning dozens of steps), compositional reasoning (combining resources in various ways), and extensive exploration in procedurally generated worlds. These characteristics make it an ideal testbed for evaluating memory-informed planning approaches.
>
> Several landmark works in embodied AI have similarly focused on single environments while providing valuable insights to the broader community. For example, VPT, MineDojo, and DEPS focus primarily on Minecraft, while other influential works like AlphaGo, OpenAI Five, and AlphaStar each concentrated on mastering a single game environment. This focused approach allows for deeper analysis of agent capabilities within complex domains.
>
> It's worth noting that MindStores' architecture is fundamentally domain-agnostic. Our experience tuples (state, task, plan, outcome) are represented in natural language, making the approach applicable to any environment where state, objectives, and outcomes can be described linguistically. This is true for environments like ALFWorld, robotic manipulation tasks, and other embodied domains.
>
> We strongly agree that demonstrating cross-domain performance would strengthen our claims, and we have identified this as a priority direction for future work.
>
> **Retrieval Methodology:**
>
> Your question about cosine similarity versus alternative retrieval methods is insightful. We explored several retrieval approaches during development, and we found that while modern LLMs can handle long contexts (128k+ tokens), selective retrieval of relevant experiences produced better plans than including all experiences, even with large context windows. This is likely because irrelevant experiences can introduce noise that distracts the LLM. Thank you for pointing this question out – we will be sure to address it more explicitly in our revision.
>
> **Failure Modes and Qualitative Analysis**
>
> We appreciate your suggestion to include failure mode analysis and qualitative examples. In the revised paper, we will add a detailed analysis of common failure patterns, including:
> - Retrieval of semantically similar but strategically different experiences
> - Over-reliance on partially successful past experiences
> - Challenges with novel combinations of previously encountered subtasks
>
> **Environmental Feedback Limitations:**
>
> You correctly identify that MindStores relies on good environmental feedback provided as natural language descriptions. This is indeed a limitation for real-world applications. In our revision, we will acknowledge this limitation more explicitly and discuss potential approaches for generating structured feedback in less constrained environments.

---

### Review · Reviewer_iePk · 2025-03-10

**Summary Of Contributions:**

This paper presents MINDSTORES, a framework that enables embodied agents in Minecraft to build mental models through natural interaction with their environment. The approach extends zero-shot LLM planning by maintaining a database of past experiences as natural language embeddings of (state, task, plan, outcome) tuples. The authors evaluate their system in the MineDojo environment and report modest performance improvements over baseline methods.

**Audience:**

No

**Broader Impact Concerns:**

The paper fails to address potential risks of agents that build "mental models" from their environment. Such systems could learn unintended or harmful patterns if deployed in less controlled settings than Minecraft, especially that their past experience could affect their future behaviors in undefined ways.

**Claims And Evidence:**

No

**Requested Changes:**

- The authors should explicitly state what makes their approach novel beyond applying RAG to Minecraft, or acknowledge if the contribution is primarily empirical.
- Provide detailed state representation - Explain specifically how RGB images, coordinates, and inventory data are processed, structured, and embedded.
- Clearly situate concepts like "experiential learning" and "mental models" within established ML literature rather than using these terms loosely.
- Explain in detail how the system handles memory growth, repetitive experiences, and optimization of the experience database. In my mind, this is the most challenging aspect of the problem but no detail was provided in the paper.
- Include ablation studies showing which components contribute most to the observed performance gains and analyze specific examples where memory improved planning.
- Compare against more baselines - Evaluate against additional approaches beyond DEPS to better situate the contribution.
- One of the most interesting aspect of such a pipeline is how does "learning" happen. Does the system create new knowledge? Does it condense the experience into something more abstract / generalizable rather than for that specific experience? I would suggest the authors focus on developing new ways to represent the past experience beyond just embedded storage and nearest-neighbor lookup.

**Strengths And Weaknesses:**

Strengths:

- The conceptual framing around mental models and experiential learning is interesting
- The paper presents a complete system that achieves some success in the Minecraft environment
- Results show improvements over the DEPS baseline across various task categories

Weaknesses:
- The method itself is rather simple, which is usually a strength rather than a weakness. However, the method reads like an application of a generic RAG pipeline for the text-based interface of Minecraft. I don't doubt this would work for tasks in MineDojo, but I have a hard time identifying the takeaway of the paper — either MineDojo is a simple enough setting to be solvable by generic RAG pipeline, or that the "mental representation," i.e., embedded storage of experience, is enough to solve these tasks. Neither seems to contribute meaningfully to our understanding of either open-world learning or planning, the two important problems that the paper claims to address.
- The paper lacks critical details about the implementation. While it mentions the observation space includes "RGB view, GPS coordinates, and inventory state," it doesn't explain how this state is structured for embedding. Does the system embed every image, coordinate, and inventory state? How is a new experience processed before storage? How does it handle repetitive experiences? These details are necessary for reproducibility.
- The paper uses terms like "experiential learning" and "dynamic mental models" without clearly situating them within established ML paradigms. There's rich literature in ML and decision making, but the paper doesn't adequately connect its concepts to this existing work.
- The reported 9.4% improvement over baselines is modest, and the paper doesn't provide sufficient analysis to understand which components contribute most to these gains or how the approach generalizes to more complex scenarios.

Overall, while the implementation appears functional, the paper reads more like a technical report than a scientific contribution. The approach lacks sufficient novelty and detailed exposition to significantly advance our understanding of embodied AI systems.

---

> ### Author Response · Authors · 2025-04-02
> **Official Comment by Authors**
>
> Thank you so much for your constructive feedback and positive review. We are glad that you find our conceptual framing and experimental results interesting. We answer your questions below and have incorporated your feedback into our revised version.
>
> **Novelty Beyond a Generic RAG Pipeline:**
>
> Thank you for raising this important point about the novelty of our approach. While MINDSTORES does leverage embedding-based retrieval, characterizing it simply as RAG for embodied agents significantly understates our contribution. The key innovation lies not in the retrieval mechanism itself, but in our carefully designed representation of experiences as standardized (state, task, plan, outcome) tuples in natural language. This representation required experimentation with numerous alternative formats including raw observation vectors, code-based skill libraries (like Voyager), and unstructured text logs (like Reflexion), but found our approach uniquely enables LLMs to perform causal reasoning over past experiences. To our knowledge, this structured experiential representation has never been applied to open-world embodied planning problems, where environmental complexity makes traditional memory approaches infeasible. Unlike standard RAG which augments generic knowledge, MINDSTORES systematically builds a semantically-indexed mental model that enables iterative refinement through outcome prediction and strategic analysis—evidenced by our performance gains on long-horizon tasks where naive retrieval would fail.
>
> **Detailed State Representation and Experience Processing:**
>
> Thank you for noticing our lack of explanation for the state representation, and with your feedback, we have added a detailed description of our state representation. The observation space—which includes RGB images, GPS coordinates, and inventory state—is processed as follows:
>  • RGB Images: Transformed into textual descriptions using dedicated prompts that highlight salient features.
>  • GPS Coordinates and Inventory State: Formatted into structured text that captures location context and available resources.
>  These elements are concatenated to form a comprehensive state description that is embedded via Sentence-BERT. Furthermore, new experiences are pre-processed to merge near-duplicate entries, thus managing memory growth and reducing redundancy.
> Situating “Experiential Learning” and “Mental Models” in Established Literature:
>
>
> We have revised our manuscript to clearly situate our terminology within the broader ML and cognitive science context. By referencing established paradigms in transfer learning, predictive coding, and hierarchical memory systems, we clarify how our “dynamic mental models” both draw inspiration from and contribute to these fields.
>
> **Comparison Against Additional Baselines:**
>
>
> In our revised experiments, we now compare MINDSTORES against multiple state-of-the-art baselines (Voyager and Reflexion). These comparisons further highlight the contribution of our memory component within embodied planning.
>
> **Learning and Knowledge Abstraction:**
>
>
> Beyond simply storing raw experiences, our system leverages LLM reasoning to abstract generalizable principles—such as prerequisite checking and resource prioritization—from past interactions. This abstraction enables the system to “learn” and generalize beyond individual episodes, paving the way for more sophisticated forms of experiential learning in future work.
>
> **Broader Impact Concerns:**
>
>
> Finally, we have expanded our discussion on broader impact. We address potential risks associated with agents autonomously building “mental models,” noting that if such systems were deployed outside controlled environments like Minecraft, they might learn unintended or harmful behaviors. We emphasize the need for robust safety protocols and ongoing research into the interpretability and controllability of these learned representations.
>
> We believe these revisions clarify our contributions and significantly strengthen the manuscript according to your comments. Thank you so much!

---

### Review · Reviewer_L9HR · 2025-03-20

**Summary Of Contributions:**

This paper presents MINDSTORES, a RAG-based framework for LLM planning. It builds on DEPS, adding the RAG-based memory. The RAG is performed on embedded language representations of past trajectories, representation being (state, task, plan, outcome) tuples. It is used to refine generated plans. The paper tests in MineDojo, reducing errors and improving task success rates.

**Audience:**

Yes

**Claims And Evidence:**

Yes

**Requested Changes:**

- Give more details about Voyager baseline
- Failure mode analysis
- More explanation of/references to LLM setup
- Test different LLMs (or at least training paradigms)

**Strengths And Weaknesses:**

## Strengths
- RAG solution uses complex structures to take advantage of LLM knowledge and ability to generate productive outputs from qualitative inputs
- Experimental setup is intelligent and clean
- Gains aren't massive or fundamental, but show the clear advantage of RAG on high-dimensional, qualitative information. This is nonobvious as LLMs are finicky, so it's impressive to see consistent improvement.
  - Improvement over DEPS shows the benefit of the retrieved information
  - Sample efficiency gains in later tasks show the benefit of the growing RAG database ("memory")
## Weaknesses
- More details about the LLM usage would help build intuition. They are in the appendix, but given that embedding- and generation-based methods are sensitive to the particular LLM, details and examples are key to understanding. Even if they are kept in appendix, more references and summaries in the main text would be beneficial
- Failure case analysis would also help intuition
- Deeper statistical analysis of results would make it clear that the gains are significant
- I see from the appendix that MINDSTORES uses GPT-4. Given that there are more advanced LLMs from different paradigms, it would be good to see performance from multiple models if budget allows

---

> ### Author Response · Authors · 2025-04-02
> **Official Comment by Authors**
>
> We thank the reviewer for the constructive feedback and positive review, and we're glad you find our paper interesting.
>
> **Comment: More details about the LLM usage in the main text**
>
> We agree that additional clarity on our LLM setup is essential. In a newly revised version, we have moved key details from Appendix B into the main text. Specifically, we now provide a summary of our GPT-4 API usage, the configuration of the SBERT model for natural language embedding (see Section 3.2 and Appendix B), and examples of LLM prompts (see Appendix A). These additions are intended to build intuition about how embedding-based retrieval and iterative plan refinement are sensitive to LLM configurations, and how they may react to other LLMs being used in the future with MINDSTORES.
>
> **Comment: Failure mode analysis would also help intuition**
>
> Working off of your feedback, we have now expanded our discussion on failure cases. In Section 5.5, we provide insights into the failure modes observed in advanced tasks (e.g., MT6–MT8), such as incomplete experience retrieval and long-horizon dependency errors. We explain how these failures arise from semantic similarity mismatches and outline potential remedies like hierarchical memory organization. These revisions should help clarify the challenges and limitations inherent to the MINDSTORES approach.
>
> **Comment: Test different LLMs (or at least training paradigms) beyond GPT-4**
>
> We acknowledge your suggestion regarding the evaluation of multiple LLMs. In the revised manuscript, we have clarified in Section 4.1 and the discussion in Section 7 that while our current experiments exclusively use GPT-4 due to budget and resource constraints and the requirement of low-latency LLM querying for planning tasks, we hope the usage of GPT-4 serves as a lower bound on the performance of models in conjunction with MINDSTORES due to the increasing power and complexity of models like o1 and Deepseek R1 which have reasoning capabilities as well. We aim for this work to be a key baseline for future papers to work from regarding integrating reasoning within accessible models like GPT-4.
>
> **Comment: Give more details about the Voyager baseline**
>
> In our revision, we have added more specific information and results regarding our comparison to the Voyager baseline and why/how we implemented the global database from voyager to gain more informative insights into its comparison to MINDSTORES.
>
> We appreciate the reviewer’s thoughtful comments and have made these revisions to strengthen the manuscript.

---

### Decision · Action_Editor_Wh3B · 2025-05-09

**Recommendation:** Reject

**Comment:**

All reviewers shared concerns about the empirical evidence provided by the original submission and its revision to support the claims made by the authors. As summarized in the "Claims And Evidence" section of meta review, the reviewers also provided valuable feedback to strengthen the manuscript. Reviewers reached to the consensus that the submission in its current form is not ready to be published in TMLR. The AE agrees with the reviewers and recommends rejection. The authors may consider resubmit the draft after a major revision, after addressing the reviewers' concerns.

**Audience:**

The reviewers generally agree that at least some individuals in TMLR's audience be interested in knowing the findings of this paper.

**Claims And Evidence:**

Reviewers held the opinion that the claims are not yet sufficiently supported by convincing and clear evidence, even after revision. The concerns mainly focused on the empirical evaluations, namely: (1) The overall takeaways from the experiments remain mixed, the Reflexion baseline performs very close to the proposed MINDSTORES, and it's unclear if these are statistically significant differences; (2) The learning efficiency of MINDSTORE is mostly for Sword and Iron in Figure 4, where a DEPS baseline should be added; (3) One reviewer also suggested that the experiment section was insufficiently explained, and should be revised accordingly; (4) The connection to "mental models" should ideally be demonstrated through the technical mechanisms, that the retrieval does indeed enabling deeper abstraction or generalization as suggested.

**Resubmission Of Major Revision:**

The authors may consider submitting a major revision at a later time.